# Moral Judgement along the Academic Training

**DOI:** 10.3390/ijerph19010010

**Published:** 2021-12-21

**Authors:** Giulia D’Aurizio, Fabrizio Santoboni, Francesca Pistoia, Laura Mandolesi, Giuseppe Curcio

**Affiliations:** 1Department of Biotechnological and Applied Clinical Sciences, University of L’Aquila, 67100 L’Aquila, Italy; giulia.daurizio@univaq.it (G.D.); francesca.pistoia@univaq.it (F.P.); 2Department of Management, Sapienza University of Rome, 00185 Roma, Italy; fabrizio.santoboni@uniroma1.it; 3Department of Humanities Studies, University Federico II, 80138 Napoli, Italy; laura.mandolesi@unina.it

**Keywords:** moral dilemma, decision making, ethical judgement, university training, executive function

## Abstract

Moral reasoning and consequent decision making are central in the everyday life of all people, independent of their profession. It is undoubtedly crucial in the so-called “helping professions”, when the professional through his/her decisions can support or not support others. Our study aimed to investigate whether academic training can play an essential role in influencing moral reasoning. We used three different conditions: 20 moral personal, 20 moral impersonal, and 20 nonmoral dilemmas to assessed differences in moral judgement between students of Economics, Medicine, and Psychology at their first year and at the end of university training. We observed a difference between school and year of course: psychology students showing more time when asked to read and answer the proposed questions. Moreover, medical students showed a significant increase in sensitiveness to moral issues as a function of academic ageing, whereas such a moral sense regressed from the first to the fifth year of academic training in other students. Gender was also relevant, with women showing an increased response and reading times compared to than men when asked to cope with moral decisions. This study shows that the main factor driving moral decision making is the faculty to which one is enrolled, significantly modulated by sex and academic seniority.

## 1. Introduction

Debates on moral nature have occupied the center of discussions among theologians, philosophers, and laymen for many centuries [1]. This is not surprising, as morality plays a central role in the constitution of human nature. Very often, in fact, people risk material resources or even their own physical integrity to help and/or to punish perfect strangers. This happens simply to experience a sense of fairness, concern for others, and observance of cultural, social, or religious norms [2,3]. As stressed by Moll et al. [1], this inclination can go far beyond the interpersonal sphere, as humans can engage themselves in costly behaviors in order to support abstract causes, beliefs, and ideologies. The so-called “moral sensitivity” arises from a sophisticated integration of cognitive, emotional, and motivational mechanisms, internalized through an active process of cultural learning during sensitive periods of personal and individual development [4,5].

Defining morality is a difficult task, as any definition will suffer from limitations, especially when evaluated by scholars from different fields and with different cultural and theoretical backgrounds. Generally, under the operational point of view, one can define morality as the set of customs and values that are embraced by a cultural group to guide social conduct [6]. This definition can sustain a cognitive vision of morality because: (a) it implicitly accepts the existence of cultural variability of values, rules, and norms; (b) as claimed by Haidt and Graham [7], it is compatible with the role of multiple psychological domains in moral cognition (care, harm, fairness, disgust, authority); (c) it emphasizes the fact that morality, biologically speaking, is fundamentally tied to evaluation, and (d) morality emerged from our evolutionary history, probably by way of gene-culture coevolution, by means of sophisticated forms of cooperation, cohesion, and reciprocity [8].

Making moral choices and thus interpreting ideas of actions potentially interfering with others’ life as “right” or “wrong” is a process based on “moral motivations”. Motivations depend on the representation of complex moral sentiments and values, leading to a simple categorization of moral actions [1]: 1. Self-serving actions that do not affect others; 2. Self-serving actions that negatively affect others (“selfishness”); 3. Actions that are beneficial to others, with a high probability of reciprocation (“reciprocal altruism”); 4. Actions that are beneficial to others, with no direct personal benefits (material or reputation gains) and no expected reciprocation (“genuine altruism”), that includes altruistic helping as well as costly punishment of norm violators (“altruistic punishment”). Usually, we can affirm that ordinary behaviors of social mammals fall into categories 1, 2, and 3, whereas genuine altruism is mainly a human attribute [8]. Although genuine altruism is costly to the individual and is less likely with increasing cost, it benefits the survival of a social group and, therefore, may have conveyed evolutionary advantages [9].

Altruistic choices underlie prosocial acts, such as costly helping, as well as costly punishment, in which one sacrifices one’s own resources to punish somebody who violates a social norm [8]. Understanding the nature of such inclinations is a challenging task, as these behaviors can be quite costly and do not confer clear material or survival advantages from the agent’s perspective. Although theoretical biology and experimental economics have strongly substantiated the validity of these “selfless” human behaviors [9,10,11,12], the motivational sources of altruistic inclinations have only recently started to be unveiled by neuroscience. Specifically, regarding human moral behavior, it is reasonable to assume that without the engagement of motivational mechanisms, purely rational moral prescriptions (“oughts”) could not be translated into actual behaviors.

It is now well accepted that both emotion and cognition play relevant roles in moral judgment, but it is not totally clear how they interact to produce moral thoughts and choices. Some authors believe that although emotion and cognition collaborate in these decision-making processes, they are dependent on largely separable neural systems. This point of view hypothesizes a central role of the prefrontal cortex in cognitive (rational) control and inhibitory function over the limbic (emotional) automatic responses in cases of moral conflict. These top-down processes guarantee that better decisions leading to overall “greater benefit” will be made [13]. An alternative point of view emphasizes the idea that emotion and cognition are not dissociable elements underlying moral motivations, and that such motivations are represented within corticolimbic neural assemblies [1]. Following these authors, conflicting moral decisions would not entail a conflict between emotion and cognition, but between two or more choices, which rely on cortico-limbic assemblies encoding distinct motivationally salient goals. As such, a cognitive process that is devoid of motivational salience would never be able to overcome a motivationally laden choice—even if the “rational” option would be saving dozens of lives and the “emotional” one would be eating a piece of chocolate cake. As proposed by Moll and co-authors [1], moral sentiments and values are key players in moral cognition and decision making by providing these complex motivations.

Together with all issues discussed above about moral decision making, individual human differences should also be considered. Several psychological studies showed that our cognitive, emotional, and social processes can be at least modulated by individual motivations and expectations [14]. According to the cognitive-developmental approach based on Kolberg’s ideas, the development of moral reasoning occurs through change in the proportions of three cognitive schemas used while reasoning about a moral dilemma [15]. Personal Interest is the least developed schema, which is characterized by thinking about personal gains or losses of each participant of the moral dilemma or their significant others. The following and more advanced, in terms of fairness and justice, is the Maintaining Norms schema, characterized by realization that one needs to get along with people other than friends and kin, and therefore needs rules and norms to stabilize behaviours and expectation among people who are not familiar intimates and may have different interests. Finally, the most developed moral reasoning uses Post-conventional schema, characterized by the primacy of moral criteria, appeal to shareable ideals and full reciprocity. According to the theory, individuals irreversibly progress from using mostly Personal Interest towards using mostly this cognitive schema when thinking about a moral dilemma [15,16]. The critical period of transition to the Post-conventional moral reasoning is late adolescence and young adulthood [14,16]. In this period, educational experience can play an essential role and the majority of studies confirmed the positive association between moral reasoning and higher education [17,18].

Finally, as recently observed in medical students, university education can lead to the phenomenon of the paradox of the regression of moral reasoning [19], an increase in utilitarian and personalistic decisions, and the “waste” of socially and culturally acceptable ideals. Marwell and Ames [20] and later Carter and Irons [21] come to similar results in relation to groups of students of Economics. The authors demonstrate that economists are different. In particular, they claim that the behavior of students of economics is more selfish/greedy (or less pro-social) as compared to other social groups. There are two possible interpretations of this: the first, based on the idea of self-selection, assumes that those most selfish/greedy choose to study economics rather than classical philology or other subjects; the second, based on the idea of learning, claims that an economics course makes students more selfish. Some measurements of “before and after treatment” (i.e., before and after a standard course in microeconomics) seem to support the latter interpretation based on a “learning morally harmful” idea [22].

The main aim of the present study is to assess the moral sensitivity in three groups of university students (those from the School of Economics, School of Psychology, and School of Medicine) to evaluate if people with interests in different university training courses show differences in moral behavior. The study also considers the year of study: in each university population it will consider both students in their first year and in their fifth year of study: in this way we aim to investigate the possible presence in these population of the phenomenon of the so-called ‘regression of moral judgement’, an event well described in medical students [19]. Finally, independently by training course and by year of study, the weight of religiosity as well as the sex of participants will be taken into consideration.

## 2. Materials and Methods

### 2.1. Participants

A total of 516 (age mean 22.3 ± SD 3.2) university students were included in the study: 176 from the School of Psychology (University of L’Aquila), 170 from the School of Economics (Sapienza University of Rome), and 170 from the School of Medicine (University of L’Aquila). The whole sample coming from each University course was subdivided in two subsamples: those in the first year of study and those in the fifth year. Based on the initial assessment, we excluded students who were taking longer than normal to complete their university course. More information on the sample is reported in Table 1. The data collection was done between November 2018 and September 2019, before the SARS-CoV2 pandemic. Therefore, the results are not affected by the potential psychological effects of the lockdown.

Each of them, before participating, gave their written informed consent, according to the Declaration of Helsinki; the study has been approved by the Internal Review Board of University of L’Aquila (# 44/2020).

### 2.2. Procedure

Each participant, after being fully informed about the study’ objectives, completed a customized protocol (developed with SuperLab 4.1 for Windows, Cedrus Corporation, San Pedro, CA, USA) aimed at evaluating the participant’ behavior in a situation of moral decision making. The task presented a set of different moral dilemmas originally proposed by Greene and coworkers [23,24] and recently used and validated in the Italian context [25]. In each task, participants were asked to answer to 20 moral personal, 20 moral impersonal, and 20 nonmoral dilemmas, randomly administered by the SuperLab software. The moral personal dilemmas depict scenarios in which the participant behaves in a way that inflicts harm to other human beings by means of his own explicit action: also, if this action is aimed at good and positive purposes (i.e., to save someone else), these dilemmas have a higher emotional involvement. The moral impersonal dilemmas depict scenarios in which the participants do not induce damages to others with his/her actions but behaves in a politically incorrect way (i.e., violating common and shared social rules): also, these dilemmas have a relatively high emotional involvement. Finally, nonmoral dilemmas describe scenarios with a very low emotional involvement, as they violate neither moral rules nor standards of social cohabitation and cohesion. A list of the used dilemmas was reported in a previous publication [25].

Each scenario consisted of a brief written description of the above-described fictitious dilemma: participants were asked to suggest whether the resolution of each dilemma was appropriate or inappropriate and the software recorded both the type of answer (appropriate vs. inappropriate) and the time needed to read the dilemmas (Reading time) and to respond to them (Answer time), measured in milliseconds. The type of answer, reading time, and answer time were then submitted for statistical analyses. During the whole administration, they were sitting in front of a of 17” PC screen in a sound-proof, temperature-controlled, and quiet room.

After this task each participant also filled in a paper-and-pencil psychological questionnaire to assess religiosity (Salience in Religious Commitment Scale-SRCS; Roof, Perkins, 1975) [26], as this dimension has been recognized as a possible covariate of moral judgment behavior.

### 2.3. Statistical Analyses

Time needed to read (reading time) and to answer (answer time) to the three types of dilemmas (moral personal, moral impersonal, and nonmoral) measured in milliseconds were used as dependent variables and submitted to a multivariate analysis of covariance (MANCOVA) considering sex (men, women), school (Economics, Psychology, Medicine) and year (1st, 5th) as possible predictors, and level of religiosity as a potential covariate. A similar analysis was run also for the qualitative assessment of dilemmas (type of answer) calculated as the difference between the number of times participants identified as appropriate the dilemma and the number of times in which they evaluated it as inappropriate. Thus, positive values identified a tendency toward subjective acceptability (appropriateness) of dilemmas, whereas negative values indicated a tendency toward subjective unacceptability (inappropriateness). Analyses were carried out with IBM SPSS Statistics for Macintosh, version 22.0 (IBM Corp., Armonk, NY, USA).

## 3. Results

### 3.1. Type of Answer

The MANCOVA on type of answer (appropriateness vs. inappropriateness of dilemma-stimuli) indicated a main significant effect for School (F_6,1002_ = 15.18; *p* < 0.000001), highlighting that the students of Psychology and Medicine assessed as highly inappropriate moral personal dilemmas (*p* < 0.000001), whereas the opposite was observed for nonmoral ones that were evaluated as more “acceptable” (*p* < 0.000001). No statistical difference was seen for moral impersonal dilemmas (see Figure 1).

A main significant effect was also observed for sex (F_3,500_ = 2.85; *p* = 0.04): women assessed as more inacceptable than men both moral impersonal (*p* = 0.03) and moral personal (*p* = 0.01) dilemmas.

Moreover, a significant interaction school x sex (F_6,1002_ = 3.21; *p* = 0.004) emerged, showing significantly different choices in both moral impersonal (*p* = 0.002) and moral personal (*p* = 0.04) dilemmas, particularly for female students of Medicine and Psychology. Only a trend to significance was observed for nonmoral dilemmas (*p* = 0.06; see Figure 2).

Very interestingly, religiosity played a role as a covariate in this model (F_3,500_ = 2.88; *p* = 0.04), showing statistical significance for moral impersonal (*p* = 0.01) and moral personal (*p* = 0.02) dilemmas.

### 3.2. Answer Time

The MANCOVA on answer time indicated again a main significant effect for school (F_6,1002_ = 8.62; *p* < 0.000001), highlighting that independently by other variables, students of Psychology need longer time than others to respond to the moral dilemmas. This effect was statistically relevant for all kind of moral dilemmas (*p* < 0.00001), as depicted in Figure 3.

A tendency toward the statistical significance was observed for the interaction school x year (F_6,1002_ = 1.99; *p* = 0.06), particularly for nonmoral (*p* = 0.02) and moral personal dilemmas (*p* = 0.02), whereas moral impersonal ones did not show a statistical difference. This effect indicated that the time needed to give a final response regard to the moral acceptability of those dilemmas was longer for participants in the first year with respect to those of the fifth year and this was true only for students of Psychology (post hoc: nonmoral *p* = 0.018; moral personal *p* = 0.004), whereas those of Medicine showed an opposite trend (see Figure 4).

No other main effects or interactions were statistically significant.

### 3.3. Reading Time

The MANCOVA on reading time indicated a main significant effect for school (F_6,1002_ = 3.36; *p* = 0.003), showing a significant difference in time dedicated to reading the dilemmas among students of the three different university courses. This effect was evident for all kinds of different dilemmas (nonmoral *p* < 0.00001; moral impersonal *p* = 0.007; moral personal *p* = 0.002), indicating that Psychology students dedicated more time to read the dilemmas, as depicted in Figure 5.

A significant effect was observed also for sex (F_6,1002_ = 3.63; *p* = 0.013), indicating that women tend to read the dilemmas for a longer time compared to men. This was true particularly for nonmoral (*p* = 0.008) and moral personal dilemmas (*p* = 0.005), whereas for moral impersonal ones, statistical significance was not reached (see Figure 6).

Finally, an interaction school x year (F_6,1002_ = 3.29; *p* = 0.003) was statistically significant, indicating that students of Psychology took longer time to read both moral impersonal (post hoc 1st year vs. 5th year *p* = 0.037) and moral personal dilemmas (post hoc 1st year vs. 5th year *p* = 0.021) with respect to others, showing a strong reduction of time employed in reading between the first and fifth year of course, whereas students of Economics showed differences between dilemmas but no difference between year of course. Medicine students showed an opposite trend: moral personal dilemmas needed longer time to be read and such a time appeared greater in the 5th than in the 1st year of course (post hoc *p* = 0.007). The same trend was seen also for moral impersonal dilemmas (*p* = 0.007; see Figure 7).

No other main effects or interactions were statistically significant.

## 4. Discussion

The present study investigated university students attending schools of Medicine, Economics, and Psychology, with the main aim of assessing their moral sensitivity and testing whether different cultural frameworks can affect moral behavior. As a companion goal, we investigated the role of academic seniority, by comparing in the three groups students in their first and fifth years of study: this was mainly aimed at testing the hypothesis of the regression of moral judgement, an event well known and previously described in medical students [19]. To test these aims we also took into consideration the sex of participants and self-reported religiosity.

Present results indicate a generalized effect for attended school, year of study, and sex. Level of religiosity did show only a side role, acting as a covariate when participants were asked to decide about the appropriateness of moral decisions.

More specifically, regarding the time needed to read each dilemma (Reading time), the effect for school indicated that students of Psychology seem more sensitive to these kinds of dilemmas, as they need more time to read questions. The time spent to read moral dilemmas was also modulated by sex, in the sense that women tended to dedicate longer time to the reading. Finally, the interaction school x year showed that moral sensitivity regressed from the first to the fifth year of academic training in both Psychology and Economics students, whereas an opposite trend was seen in Medicine students.

With respect to the time required to answer to the dilemmas (Answer time), the same effect for school was observed, with Psychology students more thoughtful when asked to respond to the question. Again, a slight moral regression from the first to the fifth year of academic training was recorded for Psychology students, whereas an increase in sensitivity to these issues was observed in Medicine students.

Finally, the Type of answer highlighted again a difference based on the school to which the participants belonged: students of Psychology and Medicine assessed as highly inappropriate moral personal dilemmas, whereas the opposite was observed for nonmoral ones, seen as more “acceptable”. Moreover, women assessed as more inacceptable than men all dilemmas involving moral aspects. These effects were confirmed by the interaction school x sex, with women of Medicine and Psychology perceiving those moral dilemmas are much more inappropriate than the others. The MANOVA on Type of answer also revealed a mediation effect of religiosity: more religious participants tended to assess as more inappropriate all kind of moral dilemmas.

### Implications

As follows, these results confirm some effects already known in the literature, suggesting some newly observed differences between samples under investigation [27,28].

First, the difference in the time needed to read dilemmas indicate a specific mental mindset of Psychology students when put in front of issues regarding moral aspects of life, independently by the type of dilemma (i.e., moral, or nonmoral). This could reflect a kind of personality and motivational “trait” distinguishing people specifically interested in disciplines of health care that ask to think about and cope with human aspects of life [29].

Second, the progressive increase in time of reading from nonmoral to moral impersonal and then to personal dilemmas is consistent with previous literature [24,25]. Conversely, when asked to answer to the dilemmas, this trend disappeared, leaving space for individual differences between students. Present findings may reflect a conflict between deontological rules and cognitive control of problem-solving: processes of deciding and answering take a longer time in the moral vs. nonmoral conditions because the involved emotional status is much stronger and can intensify this conflict.

Another interesting point arising from the present data is related to the possibility to identify a differential effect of both academic “ageing” and membership course. In fact, some students at the end of their university training appear much more inclined to decide in a non-utilitarian way, reading (i.e., Medicine and Economics students) and answering (i.e., Medicine and, partly, Economics students) taking more time than younger students. Curiously, an opposite trend was seen with Psychology students, who showed a less thoughtful behavior in the last year of course with respect to the first one. Strikingly, this effect seems a general consequence of academic seniority, as it is present almost in all kinds of dilemmas, even if it was more evident when participants were asked to answer moral ones. This effect seen in senior Psychology students could be explained based on the difficulty to cognitively and emotionally manage situations such as those depicted in moral dilemmas: participants who read those scenarios seem to feel an impelling urgency to answer and go ahead, in order to solve as soon as possible these very engaging requests. These results are very intriguing because they do not confirm previous data about the high utilitarian and personalistic decision in students of Economics, who have been described as carriers of a kind of “behavioral fingerprint” [20,21]. Thus, the hypothesis put forward and partially demonstrated in an Italian study [22] according to which an Economics course makes and/or attract students who are more selfish, does not seem to be confirmed by present data.

In parallel, the hypothesis previously put forward about a moral regression of Medicine students [19] is not confirmed by present data. This could depend on several reasons: the cultural difference between the studied samples; the difference between the didactic core curriculum in these two countries (in Italy, in the last decades, several universities dedicate ever more space to humanities and related issue); or the role of religiosity (not accounted for in Hren et al.’s paper).

Finally, sex differences observed in this study deserve some remarks. Here women showed to be (1) slower in reading what is morally appropriate and what is not, and (2) basically predisposed to judge as inappropriate the moral dilemmas. This finding supports the idea that women are less inclined to make utilitarian choices, trying to avoid putting others at risk of danger or harm, maybe due to the fact that they could be mainly driven by emotions, empathy, and care for others, following the so-called ethics of care, whereas men could tend to solve moral dilemmas following law and order rules, according to an “ethics of justice” [30]. However, these sex-related differences could be connected to differences in empathic ability, which make women more resistant to decisions that entail directly inflicting physical or moral pain on other individuals, despite their utilitarian value [31]. These differences could depend on different neural circuitry, hormonal influences, and the cognitive structure of women when engaged in moral decision making [25].

Nonetheless, the present study has some limitations. One is related to the comparison among only three types of academic training: also, if a career in Economics and Psychology (or Medicine) are ideally deeply different, it would be interesting to also include in the study students of “hard” sciences (Biology, Chemistry, Physics), technical courses (Informatics, Engineering), or “soft” sciences (Sociology, Philosophy). This could allow a direct test of the hypothesis that moral sensitivity [32] is the base of work-and-life choices (the idea of self-selection), or that the experience we have during our life brings some particular learning(s) that consequently orient and drive our behavior. Finally, it should be borne in mind that the three compared samples come from different universities and socio-cultural realities: comparing people living in a metropolis (i.e., Roma) and in a relatively small town (i.e., L’Aquila) could account for possible differences in the moral judgment. Moreover, a more ecological test of moral dilemmas could be developed in augmented reality, allowing one to assess the cognitive processes underlying moral decision making in real-life contexts [33,34].

The debate about moral and ethical judgment and moral decision making is very relevant also in the view of planning specific training programs, dedicated to future physicians, psychologists, and more general health care professionals [35,36,37], in handling challenging moral issues, such as those regarding end-of-life care. With respect to Economics schools, we suggest introducing classes or workshops focusing on these issues, in order to raise awareness of those students on the moral implications of their work activities. Future studies, multicentric and possibly conducted in different countries, could help to shed light on these aspects that are very relevant also in the perspective of public health care. In light of this, developing specific cognitive virtual reality training for professionals called upon to make moral choices in their work could both mitigate moral conflict and the consequent stress that derives from it [38].

## 5. Conclusions

In conclusion, the present study shows that the main factor driving moral decision making is the school in which one is enrolled, modulated by sex and academic seniority. Psychology students show a greater moral sensitivity with respect to the other two academic schools, particularly those of Economics. Nonetheless, future psychologists also show a significant moral regression along the academic course. In contrast, Medicine students show a significant sensitivity to moral issues that remains stable along the training period. Women tended to be more thoughtful when involved in moral issues, dedicating more time to read them, and are generally predisposed to judge as very inappropriate all moral dilemmas. When studying moral decision making, religiosity should be carefully considered.

## Figures and Tables

**Figure 1 ijerph-19-00010-f001:**
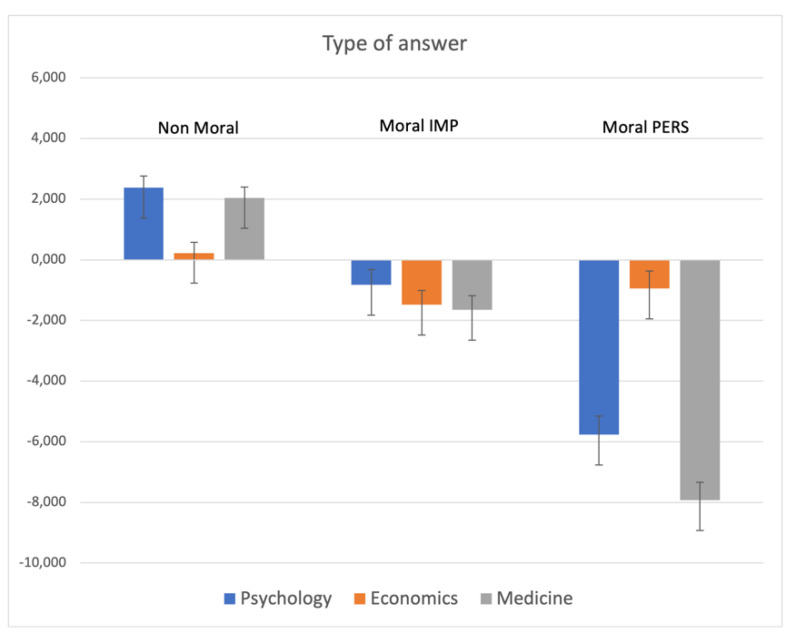
MANCOVA results on Type of Answer: significant main effect for school. Note: the y-axis shows the difference between the number of appropriate vs. inappropriate judgments.

**Figure 2 ijerph-19-00010-f002:**
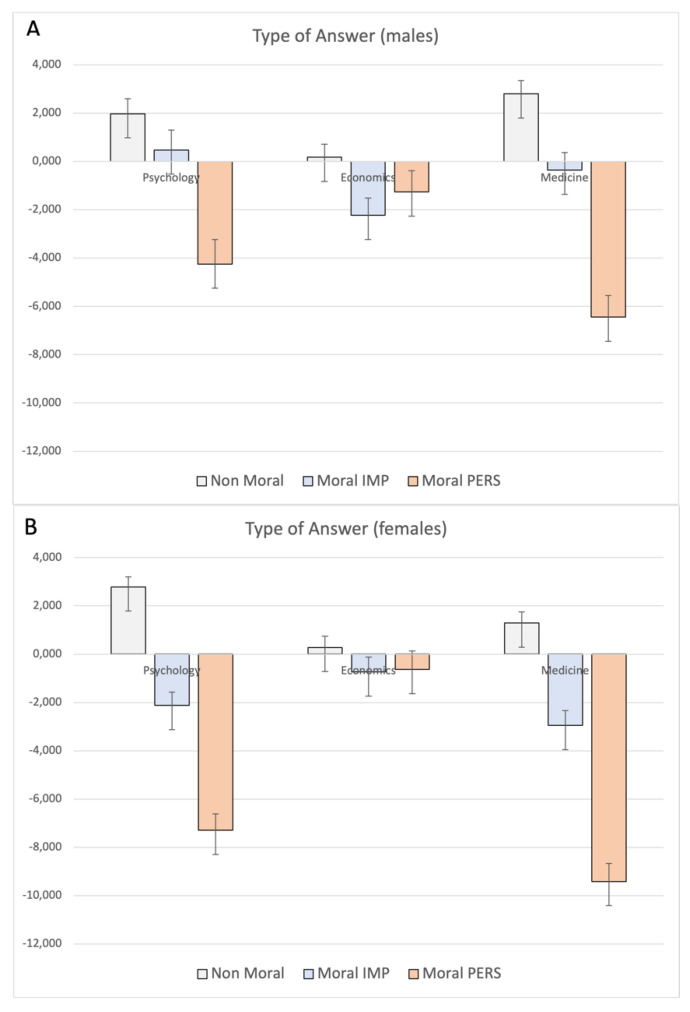
MANCOVA results on Type of Answer: significant interaction school x sex. Panel (**A**): males; panel (**B**): females. Note: the y-axis shows the difference between the number of appropriate vs. inappropriate judgments.

**Figure 3 ijerph-19-00010-f003:**
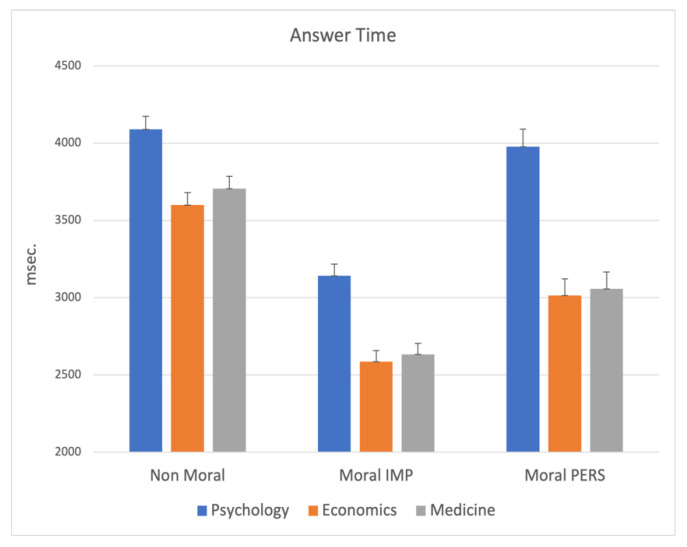
MANCOVA results on Answer Time: significant main effect for school. Note: the y-axis shows the answer time (msec.).

**Figure 4 ijerph-19-00010-f004:**
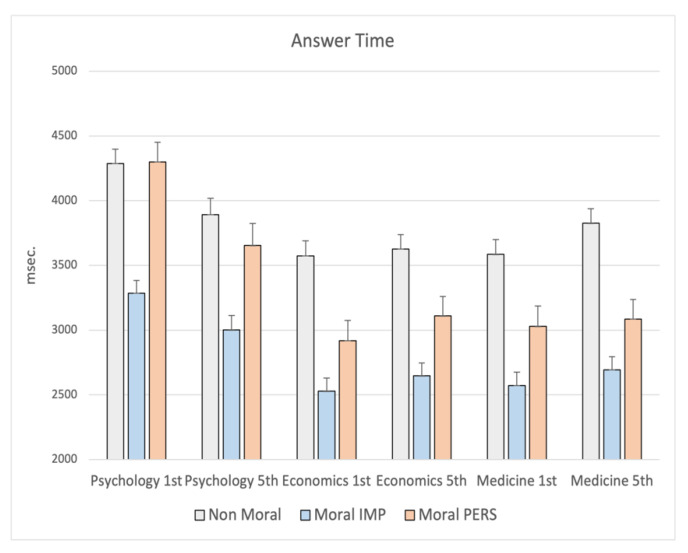
MANCOVA results on Answer Time: significant interaction school x year. Note: the y-axis shows the answer time (msec.).

**Figure 5 ijerph-19-00010-f005:**
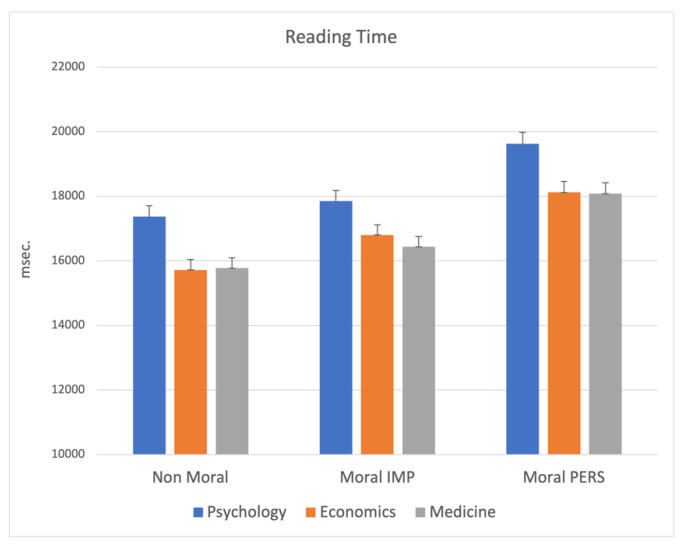
MANCOVA results on Reading Time: significant main effect for school. Note: the y-axis shows the Reading Time (msec.).

**Figure 6 ijerph-19-00010-f006:**
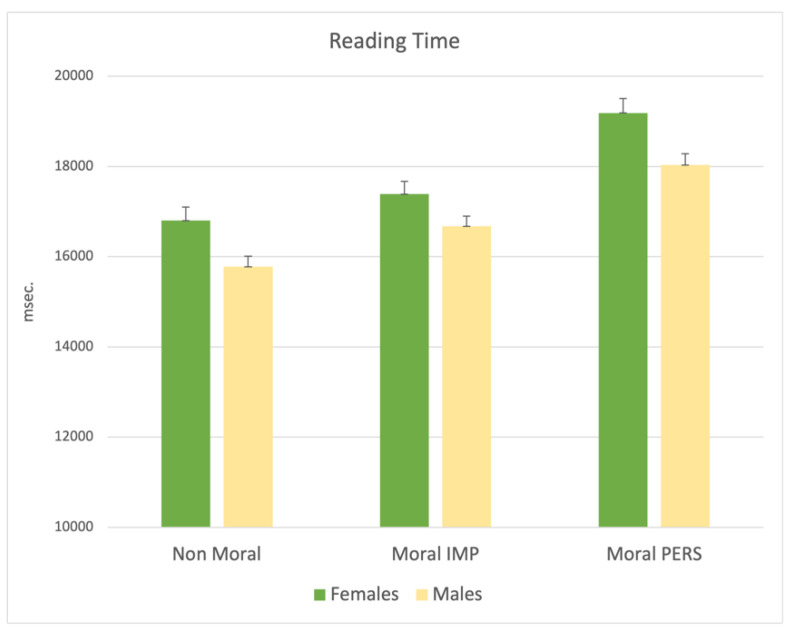
MANCOVA results on Reading Time: significant main effect for sex. Note: the y-axis shows the Reading Time (msec.).

**Figure 7 ijerph-19-00010-f007:**
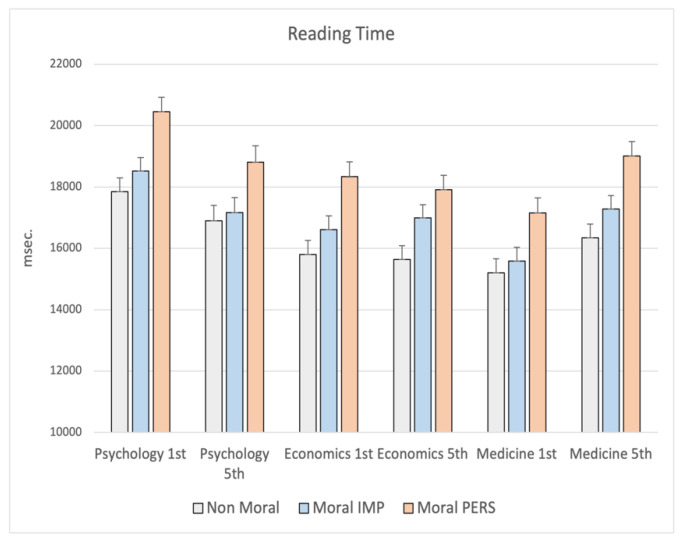
MANCOVA results on Reading Time: significant interaction school x year. Note: the y-axis shows the Reading Time (msec.).

**Table 1 ijerph-19-00010-t001:** Demographic information on the investigated sample (mean ± standard deviation).

	Total Sample (N)	Age (N) (Mean ± SD)	Sex (N)	Age (Mean ± SD)
Psychology 1st year	91	19.96 (±1.37)	M = 34 W = 57	20.74 (±1.35) 19.49 (±1.13)
Psychology 5th year	85	24.7 (±2.89)	M = 23 W = 62	25.9 (±3.8) 24.3 (±2.4)
Economics 1st year	85	19.38 (±1.15)	M = 31 W = 54	19.26 (±0.82) 19.44 (±1.31)
Economics 5th year	85	24.41 (±1.30)	M = 42 W = 43	24.58 (±1.32) 24.26(±1.27)
Medicine 1st year	85	20.47 (±1.75)	M = 32 W = 53	20.375 (±1.56) 20.53 (±1.87)
Medicine 5th year	85	25.3 (±3.52)	M = 39 W = 46	25.85 (±4.16) 24.87 (±2.83)

Note: M = male; W = women.

## Data Availability

Data is contained within the article.

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
