# Peer review of "Moral Judgement along the Academic Training"

_ijerph, 2021, doi:10.3390/ijerph19010010_

Round 1

Reviewer 1 Report

The paper scrutinizes moral judgement variations depending on academic career and seniority, by comparing the results of student at the first and fifth year of study in Economic, Psychology and Medicine. While it is already known that Economic students regress their moral sense as they progress through their academic career, this paper investigates effect of college courses and other variables (such as gender and religiosity) on moral reasoning.

The paper is relevant, useful and well written.

I believe it is necessary to add a table with the actual results (means, standard deviations) stratified by School, year of study and eventually gender.

Why do the authors use also the word “ethical” in the title, while in the paper there is only a discussion on moral reasoning?

 Personally, I think the presentation of the results would be more engaging for the readers if the authors discussed first “Type of answer”, and only after “time”. In particular because the reasons why subjects take more time are only assumptions.

It would be useful for readers if the meaning of “type of answer” is explained in concrete terms at the beginning of section 3.3 so that it is easier to process the meaning of the results.

Figure 7 is difficult to interpret. It would be more useful to split it into two graphs, one for males, one for females (or one graphic for each School, depending on which kind of comparison wants to be emphasized).

The section “discussion” is very long. It could be split in two sub-section (summary of the results and implications).

Line 219 the “t” needs to be a capital letter.

Line 220 the word “convinced” seems a judgment, I would rather use words such as “perceiving”.

I would add a paragraph with implications for universities: what should universities do to improve moral reasoning, in particular in Economics?

Best of luck with your interesting work.

Author Response

Response to Reviewer 1 Comments

Point 1: I believe it is necessary to add a table with the actual results (means, standard deviations) stratified by School, year of study and eventually gender.

Response 1:  We are very sorry, but it seems very difficult to summarize all the results for all considered variables in a single table! Just to give an esteem, the final results file of SPSS is a pdf document of 73 pages...

Point 2: Why do the authors use also the word “ethical” in the title, while in the paper there is only a discussion on moral reasoning?

Response 2: The title has been changed, as suggested by Reviewer.

Point 3: Personally, I think the presentation of the results would be more engaging for the readers if the authors discussed first “Type of answer”, and only after “time”. In particular because the reasons why subjects take more time are only assumptions.

Response 3: This suggestion has been accepted: in the current version of the manuscript, the order of presentation of the results has been changed. As a consequence, the number of figures has been re-written.

Point 4: It would be useful for readers if the meaning of “type of answer” is explained in concrete terms at the beginning of section 3.3 so that it is easier to process the meaning of the results.

Response 4: As requested, we have now explained more in depth the meaning of “type of answer”.

Point 5: Figure 7 is difficult to interpret. It would be more useful to split it into two graphs, one for males, one for females (or one graphic for each School, depending on which kind of comparison wants to be emphasized).

Response 5: We thank the Reviewer for the suggestion: we decided to split the figure into two graphs, with the aim of emphasize differences based on gender.

Point 6: The section “discussion” is very long. It could be split in two sub-section (summary of the results and implications).

Response 6: We followed the Reviewer’ suggestion, splitting the Discussion into two subsections. Since the journal does not suggest such a structure of the Discussion section, we also ask the Editor to endorse this change.

Point 7: Line 219 the “t” needs to be a capital letter.

Response 7: The suggested change has been included.

Point 8: Line 220 the word “convinced” seems a judgment, I would rather use words such as “perceiving”.

Response 8:  We agree with the Reviewer. In the current version of the manuscript, we have changed it.

Point 9: I would add a paragraph with implications for universities: what should universities do to improve moral reasoning, in particular in Economics?

Response 9: We added some possible suggestions to Economics schools in the last para of the Discussion section.

Reviewer 2 Report

The submitted manuscript presents the results of a conducted study on moral and ethical judgement along the academic training. The students of three different courses are compared with each other to draw some general conclusions. The article is, in general, quite interesting to read. Below I include some issues that have to be addressed in order to increase its quality.

  1. When this study was performed? Please add some details.

  1. In some places of the manuscript I get the feeling that the chosen faculty somehow “shapes” the students. I think this cannot be a direct conclusion, because – as correctly stated in some other parts of the paper – people tend to chose the faculty according to their personal character/temperament. So we have a correlation, and not a “cause-and-effect” relationship influenced directly by the course of study. Some parts of the text should therefore be revised.

  1. Line 70: ordering in the group of references should be changed.

  1. Line 149: duplicated word “Note”.

  1. Numbers in section 3 (Results) should all start with 0, when lower than 1, e.g. line 200: should be p=0.003 instead of p=.003.

  1. The description of y-axis is missing in Figures 1-7.

  1. Charts in figures 6 and 7 are hard readable. Maybe another type of chart should be considered to present these results?4

  1. Line 296: this short paragraph is connected with the following ones. To indicate it clearly, extend this one-sentence paragraph with a statement like “as follows”.

  1. The list of references needs to be updated to contain more recent works, especially from the last 3-4 years.

  1. English language is fine, but there are some minor typing or gramma errors. Please carefully check the whole manuscript.

Author Response

Response to Reviewer 2 Comments

Point 1: When this study was performed? Please add some details.

Response 1: The data collection was done in between November 2018 and September 2019, far from the SARS-CoV2 pandemic. Therefore, the results are not affected by the potential psychological effects of the lockdown.

Point 2: In some places of the manuscript, I get the feeling that the chosen faculty somehow “shapes” the students. I think this cannot be a direct conclusion, because – as correctly stated in some other parts of the paper – people tend to chose the faculty according to their personal character/temperament. So, we have a correlation, and not a “cause-and-effect” relationship influenced directly by the course of study. Some parts of the text should therefore be revised.

Response 2: We thank the reviewer for this suggestion. It was also our own feeling and in fact, throughout the manuscript, we tried to stress that these data are only correlational and absolutely not a cause-and-effect relationship! Since the Reviewer did not indicate the precise points of these “feelings”, we tried to change what resulted much evident to us: we hope to have adequately interpreted and answered to Reviewer’ concern.

Point 3: Line 70: ordering in the group of references should be changed.

Response 3: The requested change has been done

Point 4: Line 149: duplicated word “Note”.

Response 4: The requested change has been done

Point 5: Numbers in section 3 (Results) should all start with 0, when lower than 1, e.g. line 200: should be p=0.003 instead of p=.003.

Response 5:  The requested change has been done

Point 6: The description of y-axis is missing in Figures 1-7.

Response 6: The requested change has been done.

Point 7: Charts in figures 6 and 7 are hard readable. Maybe another type of chart should be considered to present these results?

Response 7: Both figure 6 and 7 hve been changed.

Point 8: Line 296: this short paragraph is connected with the following ones. To indicate it clearly, extend this one-sentence paragraph with a statement like “as follows”.

Response 8: The requested change has been included

Point 9: The list of references needs to be updated to contain more recent works, especially from the last 3-4 years.

Response 9:  The Reviewer is right. In the current version of the manuscript, we have included several studies published in the last 3/4 years.

Point 10: English language is fine, but there are some minor typing or gramma errors. Please carefully check the whole manuscript.

Response 10: A full revision has been done, with a careful check of the manuscript.

Round 2

Reviewer 2 Report

The authors have improved the quality of the manuscript. I am satisfied with the responses to my previous comments as well as with the changes in the manuscript.

I am glad to recommend the acceptance of the paper, after introducing some minor modifications.

Please add the information when the study was performed to the manuscript (your answer to point 1 of my previous review). I cannot find it in the current version of the manuscript.

Figure 2 – should be bigger. Right panel is females, left panel- males. The caption must be changed.

Figures 3-7 – time – if possible add the unit (msec) also for the y-axis in the charts.

Author Response

Point 1: Please add the information when the study was performed to the manuscript (your answer to point 1 of my previous review). I cannot find it in the current version of the manuscript.

Response 1: We thank the reviewer for this suggestion. The requested change has been done.

Point 2: Figure 2 – should be bigger. Right panel is females, left panel- males. The caption must be changed.

Response 2: We thank the reviewer for this suggestion. The requested change has been done. Now figure 2 contains Panel A for males and panel B for females.

Point 3: Figures 3-7 – time – if possible add the unit (msec) also for the y-axis in the charts.

Response 3:  We thank the reviewer for this suggestion. The requested change has been done.